# SPEAKS study: study protocol of a multisite feasibility trial of the Specialist Psychotherapy with Emotion for Anorexia in Kent and Sussex (SPEAKS) intervention for outpatients with anorexia nervosa or otherwise specified feeding and eating disorders, anorexia nervosa type

Anna Oldershaw [1,2] Tony Lavender,[2] Randeep Basra,[1] Helen Startup[3]

For numbered affiliations see end of article.

**Correspondence to**
Dr Anna Oldershaw;
annaoldershaw@hotmail.com

## ABSTRACT

**Introduction** Anorexia nervosa (AN) is a severe mental health condition associated with high mortality rates and significantly impaired quality of life. National guidelines outline psychotherapeutic interventions as treatments of choice for adults with AN, but outcomes are limited and therapy drop-out high, resulting in calls for new innovative treatments. The Specialist Psychotherapy with Emotion for Anorexia in Kent and Sussex (SPEAKS) research programme sought to develop the SPEAKS intervention avoiding some difficulties inherent in development of earlier interventions, such unclear hypotheses about change processes. SPEAKS focuses on a core hypothesised maintaining factor (emotional experience) with clear proposed model of change. The current feasibility trial aims to provide an initial test of SPEAKS and inform design of a full randomised controlled trial protocol.

**Methods and analysis** This study employs a multisite, single-arm, within-group, mixed-methods design. Up to 60 participants (36 therapy completers) meeting inclusion criteria will be offered the SPEAKS intervention instead of treatment-as-usual (TAU). SPEAKS is a weekly psychotherapy lasting nine to 12 months, provided by trained and experienced eating disorders therapists. All other clinical input remains inline with TAU. Acceptability will be assessed using VAS scales and end of therapy interview. Reach and recruitment, such as recruitment yield, will be monitored. To support sample size estimation and economic estimation, data pertaining to eating disorder-related symptoms will be recorded every 3 months, alongside service usage and intervention-specific measures. Videoed therapy sessions will inform model adherence. Additional analyses coding videoed therapy will test SPEAKS change process hypotheses.

**Ethics and dissemination** Ethical approval has been granted by London–Bromley Research Ethics Committee (NHS Rec Reference: 19/LO/1530). Data will be disseminated via high-impact, peer-reviewed journals

## Strengths and limitations of this study

► This feasibility trial studies a novel psychotherapy for adults with anorexia nervosa (AN), with a focus on emotional change: The Specialist Psychotherapy with Emotion for Anorexia in Kent and Sussex (SPEAKS) intervention.
► This feasibility study is strengthened by its multi-site trial design, completed in largely 'research naïve' National Health Service services, increasing insight into its feasibility and generalisability.
► The study is limited by its single arm design and non-blinded procedures, which may result in biases such as allegiance effects and does not allow for estimating acceptability of a randomised design.
► In addition to its feasibility aims regarding the SPEAKS intervention, the trial design includes analysis of the therapeutic change process with wider implications for understanding therapeutic change for adults with AN.
► The trial has been subject to the challenges imposed by the COVID-19 pandemic including regarding clinical and research engagement.

(Open Access preferred), conferences, service user and charity networks (eg, UK charity BEAT) and through a free open conference hosted by National Health Service Trusts and Higher Education Institutions.
**Trial registration number** ISRCTN11778891.
**Trial status** Recruitment began on 12 December 2019 and ends on 28 February 2021. All data will be collected and the trial ended by 28 February 2022.
**Protocol version** SPEAKS protocol V.3.0 (30 August 2020). Changes were made to the original protocol due to the COVID-19 pandemic. A further set of changes were

made to incorporate the measures of change processes, resulting in this being the third version of the protocol.

## INTRODUCTION
### Background and rationale

Anorexia nervosa (AN) is a severe mental illness with poor prognosis and the highest mortality rate of any psychiatric disorder.[1] Early intervention is key, but is impeded by a delay of around 18 months from symptoms emerging to treatment, and with multiple relapses likely.[2] Caregiving for somebody with severe AN is almost twice as lengthy as for somebody with a physical health disorder (eg, cancer) or other mental health difficulty (eg, psychosis).[3] Thus significant costs exist financially and emotionally to the individual, family and carers, as well as society as a whole, offering a 'compelling case for change' in services and treatment (PricewaterhouseCoopers, p.9).[2]

Most recent NICE guidelines in the UK recommend that psychotherapeutic interventions be considered for adults with AN.[4] Yet results from randomised controlled trials indicate that outpatient treatments do not outperform each other or control comparisons[5–9] with weight-restoration achieved by 20% of patients after 1 year.[7–9] AN is highly valued by sufferers resulting in poor treatment engagement and high drop-out rates.[10] Thus there is an urgent need to develop innovative interventions that can engage people with AN,[4 11 12] while targeting unique factors involved in its development and maintenance.[13] Emotional experience has long been recognised as a factor in the development and maintenance of AN and is recognised as a promising target area for therapies[14] and increasingly incorporated into psychotherapy interventions for adults with AN.[5–9] However, outcomes remain limited and it is unclear to what extent emotional difficulties are targeted by these interventions.[15]

This trial comprises part of the Specialist Psychotherapy with Emotion for Anorexia in Kent and Sussex (SPEAKS) programme which aimed to develop and test in a feasibility study an emotion focused intervention for adults with anorexia (the SPEAKS intervention). This research programme sought to overcome several key difficulties with the development and application of some earlier interventions, such as unsuccessful targeting of variables or lack of clarity on how change is achieved. The SPEAKS programme proposes focussing on a core clearly defined model with one key putative maintaining process (emotional experience in AN) and following an 'interventionist-causal model approach'.[16] By developing a new model of the development and maintenance of AN drawn from integration of quantitative and qualitative research, it seeks to ensure an emotions' focus.[17] The SPEAKS intervention hypothesises a clear and testable change process to be targeted in therapy (figure 1)[18]; thus affording the ability to examine proposed mechanisms of change enabling further evidence-based development and refinement of the model. This body of work involved close partnering with stakeholders including those with

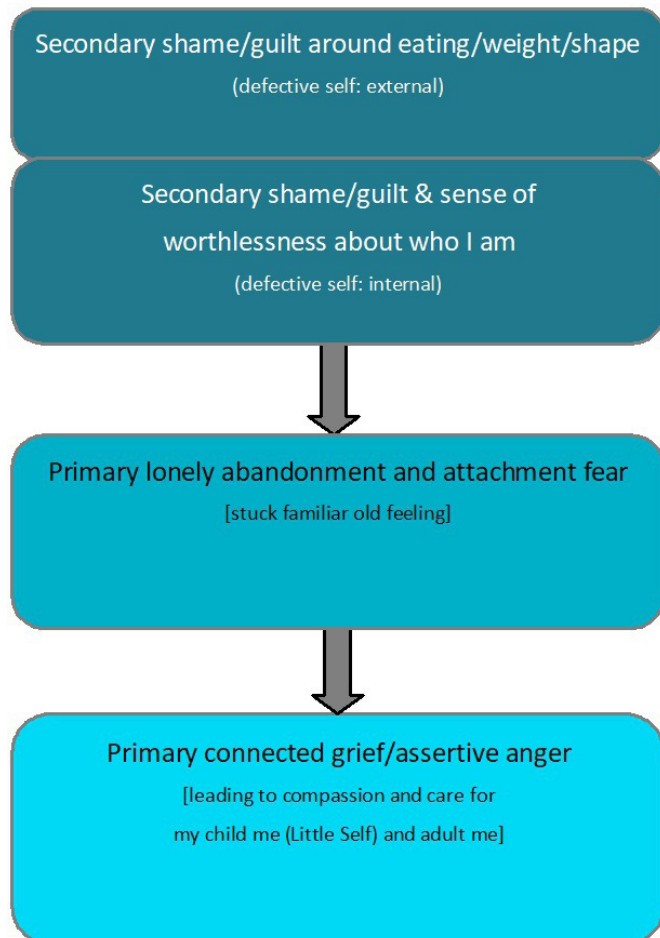

**Figure 1** SPEAKS emotion change process. SPEAKS, Specialist Psychotherapy with Emotion for Anorexia in Kent and Sussex.

current and past experience of AN, families, therapists and service managers.

### Objectives

This multisite feasibility trial aims to investigate the SPEAKS intervention in the following domains:
► Acceptability.
► Reach and recruitment.
► Adherence and compliance.
► Sample size and economic evaluation to establish parameters and financial costs of a potential future efficacy/effectiveness trial.
► Change process analysis.

We hypothesise that:
1. SPEAKS will be acceptable to participants and therapists.
2. SPEAKS will meet sufficient reach and recruitment expectations to support progression to full effectiveness/efficacy trial.
3. Therapists and service users will be able to adhere to the therapy model as intended and the research requirements
4. Participant change over time will reflect the proposed emotion change process outlined in figure 1.

## Trial design

The SPEAKS study is a multisite, single-armed, within-group mixed-methods design.

## METHODS: PARTICIPANTS, INTERVENTIONS AND OUTCOMES
### Study setting

This feasibility study runs in two outpatient specialist eating disorders services (EDS) within the UK National Health Service (NHS): Kent and Medway All Age EDS at NELFT and Sussex EDS at SPFT.

### Eligibility criteria
#### Service users

Service users are eligible to participate if they:

1. Are referred into Kent or Sussex EDS and meet service criteria (eg, registered with a local general practitioner).
2. Meet Diagnostic and Statistical Manual 5 Criteria for Anorexia Nervosa or OSFED (Other Specified Feeding or Eating Disorder) of Anorexic type.
3. Are aged 18 or above.
4. Have body mass index (BMI) >15 kg/m$^2$.
5. Have sufficient English language abilities to complete a talking therapy.

Service users are excluded if they have/are:

1. Rated as 'High Risk', or as 'High Concern' in weight criteria, on the MARSIPAN Guidelines for adults with eating disorders (ie, BMI <15 kg/m$^2$; weight loss >500 g for two consecutive weeks)[19]
2. Considerable psychological risk, including active suicidal thoughts and plans.
3. Comorbidity requiring treatment priority.
4. Alcohol/substance use disorder.
5. Participating in another treatment trial
6. Diagnosed Intellectual disability impeding ability to use therapy.
7. Pregnant.

#### Therapists

Therapists are eligible to offer SPEAKS in the trial if they:

1. Are a specialist eating disorder therapis >3 years experience.
2. Work in Kent All Age EDS or Sussex EDS.
3. Have specialist training in an experiential dialogical self chairwork model (eg, emotion-focused therapy (EFT), schema therapy (ST), compassion-focused therapy).

### Interventions
#### Speaks intervention

SPEAKS is an individual outpatient psychotherapy for adults with AN. Participants receive weekly individual sessions of psychotherapy for 9–12 months with two follow-up sessions within 3 months of completion. SPEAKS is intended to be offered face to face in a clinic setting; however, due to the COVID-19 pandemic video sessions via on an online platform are provided.

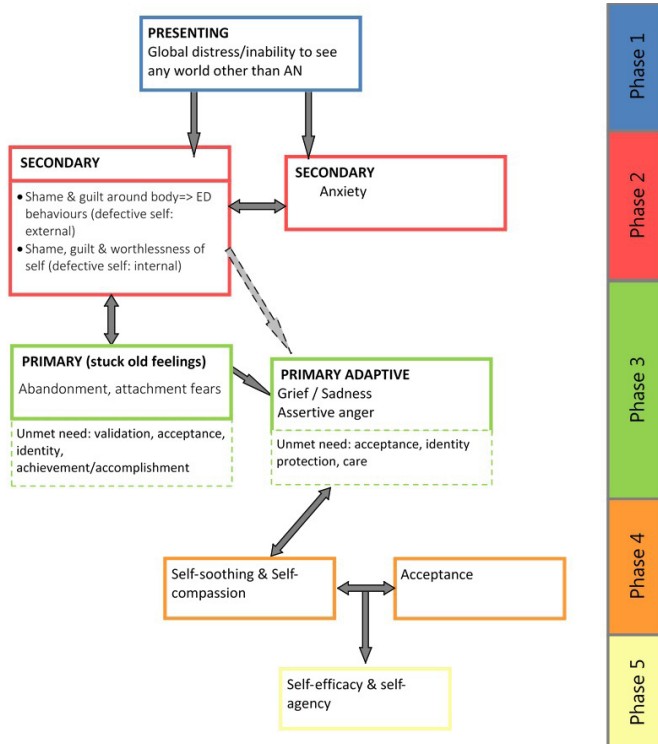

**Figure 2** SPEAKS treatment phases. AN, anorexia nervosa; SPEAKS, Specialist Psychotherapy with Emotion for Anorexia in Kent and Sussex.ED= eating disorder

SPEAKS is a direct replacement for psychotherapy as usual. Therapists receive fortnightly supervision by supervisors trained in the supervision requirements of SPEAKS. The services at both sites are in the same region of the UK and follow national NHS treatment guidelines. All usual care procedures, including additional interventions such as dietician appointments or inpatient referrals will remain the same, but be reported. People will be removed from the trial if they require immediate inpatient treatment at any point or request removal for any reason, and these data will be reported.

### Intervention development

The development of SPEAKS is consistent with MRC guidance for complex interventions.[20] SPEAKS was developed following a clear programme of research to integrate quantitative and qualitative data to achieve an initial model of the presentation (lost emotional self)[17] and potential necessary elements for therapeutic change (eg, Drinkwater *et al*, in prep). This resulted in a clear hypothesised change process to be targeted in therapy.[17] SPEAKS is organised into five phases, with associated mechanisms of change and therapeutic 'tasks' (see figure 2 for brief overview). The intervention developers (AO, TL nd HS) applied relevant psychological therapy models to target highlighted mechanisms, resulting in an integrative therapy drawing chiefly from EFT and ST. The therapy thus relies on dialogical self theory, and therapeutic tasks are those well established in EFT and ST,

such as 'chairwork' interventions to enable 'parts of self' to communicate.

### Intervention guidebook

The SPEAKS intervention is written up in a guidebook for therapists to follow. It outlines SPEAKS change process, mapping it onto expected 'phases' of therapy ([figure 2]). In each phase, associated hypothesised mechanisms are outlined and 'therapeutic tasks' described. The guidebook meets requirements for preliminary feasibility evaluation as outlined in the stage model approach for developing a psychotherapy[21] and assists intervention adherence.

### Therapist training and supervision

In additional to background training in an experiential dialogical self model, therapists receive 4 days SPEAKS model and therapy training, with further regular training days organised throughout the trial. Ongoing fortnightly supervision is delivered by SPEAKS developers. Where participants consent, all therapy sessions are video recorded for supervision. Regular reflective practice groups are offered by a SPEAKS developer and psychologist external to the clinical services (TL).

### Treatment fidelity, untoward events and protocol adherence

Reviewing videotaped sessions in supervision ensures competent treatment delivery adhering to the SPEAKS model. This is usual good practice in both EFT and ST supervision models. Case records of each therapy session in client notes outline the phase of therapy, emotions focused on and therapeutic tasks employed.

Usual service standard operating procedures (SOPs) regarding clinical care and risk management will be followed. Any protocol violations impacting delivery of SPEAKS such as admission to hospital will be recorded as per Trust guidance, and also logged in trial records for later reporting. Participants who are admitted to hospital will be withdrawn from the study.

### Outcomes
### Measures of acceptability

Triangulation of qualitative and quantitative data will address acceptability of SPEAKS to participants and therapists.

### Qualitative

Participant and therapist lived experience of SPEAKS will be examined using post-intervention semi-structured interviews adapted from the Client Change Interview,[22] completed by the researcher. Acceptability of aspects of the current and future trial designs are also addressed including use of questionnaires and willingness to be randomised to treatment arms.

### Quantitative

Acceptability and perceived value of core SPEAKS components will be quantitatively assessed using visual analogue scales created for the study using questions based on the acceptability interview, and rated on a seven-point scale strongly agree to strongly disagree (eg, I think the focus of SPEAKS makes sense for me and my difficulties). Numbers of people who choose to end their therapy because they do not think it is acceptable will also be calculated.

### Measures of reach and recruitment

Screening of potential participants will be logged, and details of unmet inclusion criteria anonymously recorded. Recruitment yield will be monitored. Where participation is declined, reasons will be anonymously recorded. Completeness of measures at each time point will be reported.

### Measures of adherence and compliance

Treatment fidelity strategies will be employed consistent with the treatment fidelity checklist,[23] including a clear intervention description (guide-book) and standardised therapist training. As described, session recordings will enable assessment of adherence to SPEAKS, where consented to. A random selection of therapy tapes will be assessed for fidelity and adherence to the model judged by the intervention developers. They will assess core components of the treatments such as empathy (using the Therapist Empathy Scale[24]), appropriate task selection based on identified tasks markers and appropriate task resolution. Data regarding treatment length, session numbers and content will be reported.

### Measures for sample size estimation

Clinical outcomes measures. In line with DSM V criteria for AN and OSFED-AN type (atypical AN) which emphasises rate of weight loss and ED cognitions, the primary outcome measure is of eating disorder cognitions and behaviours for sample size estimation—Eating Disorder Examination Questionnaire (EDEQ).[25] Other indications of symptoms, such as BMI (kg/m$^2$); Depression, Anxiety and Stress Scales 21[26] and Clinical Impairment Assessment[27] are also collected.

Intervention-specific measures. Other clinical and Intervention-specific measures will also be assessed as follows: Beliefs About Emotions Questionnaire,[28] Young's Schema Questionnaire–Short Form (SF),[29] Schema Mode Inventory ED–SF (SMI-ED-SF)[30]; Silencing the Self Scale,[31] The Sense of Agency Scale[32] and Difficulties with Emotion Regulation Scale.[33]

### Measures of economic evaluation

An adapted version of the Client Sociodemographic and Services Receipt Inventory[34] will collect economic data preintervention and postintervention to assess treatment costs received in the 6 months preceding and during SPEAKS.

### Measures of SPEAKS Change Process

The following analyses will test hypothesised therapeutic change process.

1. Videoed therapy sessions will be analysed according to two coding systems to better understand change associated with better clinical outcomes, including hypotheses about both content and order of change. The Innovative Moments Coding System[35] is a systematic reliable method for identifying innovative moments (IMs) in therapy, categorising by type (action, reflection, protest, reconceptualisation, performing change). IMs are measured by the percentage of time spent elaborating on each IM (temporal salience). The Classification of Affective Meaning States (CAMS)[36] codes the presence of emotions expressed during therapy videos in) 1 min segments. The CAMS includes nine emotion categories: global distress (rejecting anger, fear/shame, negative self-evaluation, unmet need, relief, assertive anger/self-compassion, hurt/grief, and acceptance/agency) which will be analysed to assess changes in patterns of emotion expression over time including emotion types and frequency.[37]

2. Thematic analysis will be applied to psychological formulations of 'schema modes' constructed by participants during therapy as compared against those endorsed in a quantitative measure (SMI-ED[30]). This will enable us to improve the SPEAKS intervention based on the most salient schema modes, and mode changes associated with better outcomes.

Participants may participate without agreeing to their formulations or therapy recordings being analysed.

## Participant timeline

Participants will be assessed using quantitative variables examining clinical and emotion change collected preintervention, at 3, 6 and 9 months into the intervention and

| | SPEAKS STUDY PERIOD | | | | |
|---|---|---|---|---|---|
| | Enrolment | Post-enrolment | | | Post-intervention |
| TIMEPOINT | -$t_1$ | Baseline | 3 months | 6 months | 9 months | 12-15 months |
| ENROLMENT: | | | | | | |
| Eligibility screen | X | | | | | |
| Informed consent | X | | | | | |
| INTERVENTION: | | | | | | |
| SPEAKS ASSESSMENTS | | | | | | |
| Adapted CSSRI | | X | | | | X |
| Clinical outcome measures | | X | X | X | X | X |
| Intervention-specific measures | | X | X | X | X | X |
| Qualitative interviews | | | | | | X |

Figure 3 SPEAKS spirit schedule. CSSRI, Client Sociodemographic and Services Receipt Inventory; SPEAKS, Specialist Psychotherapy with Emotion for Anorexia in Kent and Sussex.

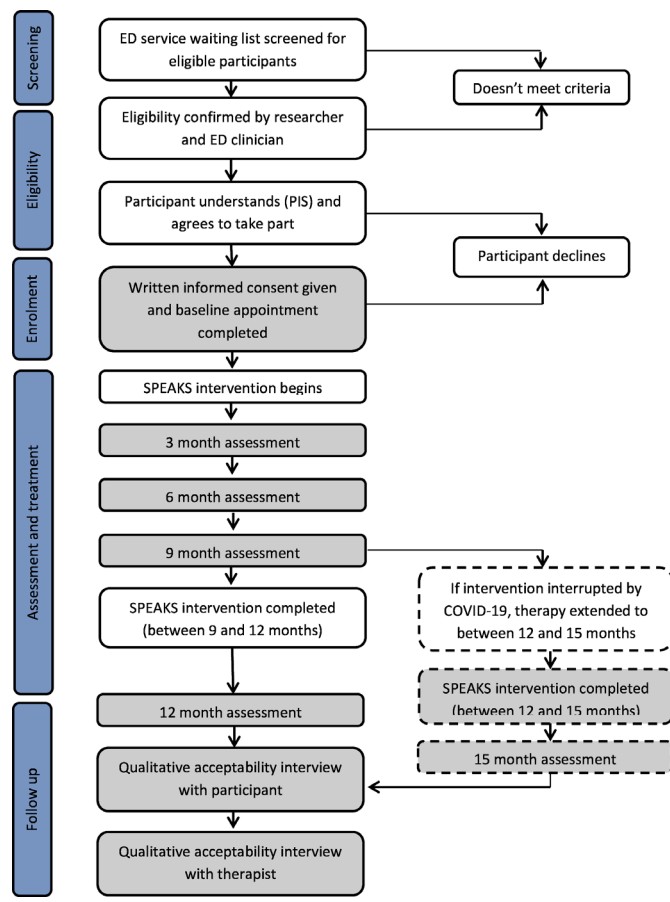

Figure 4 SPEAKS consort diagram. SPEAKS, Client Sociodemographic and Services Receipt Inventory. ED= eating disorder

postintervention (12 months; figures 3 and 4). Collecting questionnaire data every 3 months is standard practice within the EDS. Due to the coronavirus pandemic, some participants experienced a break in their SPEAKS therapy. In order to ensure that all participants are able to access the full 9–12 months of therapy intended, therapy will be extended for those impacted by a maximum of 3 months. Where relevant, this will be discussed and agreed between therapist and participant. In order to capture end of therapy data for those affected, we will include an extra data collection time point at 15 months just for these participants. The 15 months data collection point will be considered equivalent to the 12 months time point of unaffected participants, with other assessment time points adjusted accordingly to ensure progress is captured every 3 months within therapy.

Qualitative acceptability interviews with service users will be completed at the end of each participant's involvement in the study (at 12 or 15 months follow-up). Therapist acceptability interviews will be completed when they have finished working with their final SPEAKS participant.

## Sample size

The feasibility design must balance precision with unethical exposure of participants to the risks being monitored alongside unnecessary expense.[38] Teare et al[39] recommend

35 participants for sufficient feasibility data and precision of mean and variance. Therapy attrition rates for people with AN can reach 40%.[9] With a sample size of 36, we will be able to estimate a drop-out rate of 40% to within a 95% CI of ±16%. These data suggest an approximate revised sample size of up to 60 participants in order to achieve a sample of 36 participants completing therapy.

### Recruitment

Patients will be consecutive referrals meeting inclusion/exclusion criteria to the two sites in South-East England: North East London NHS Foundation Trust (NELFT) and Sussex Partnership NHS Foundation Trust (SPFT). Participants will be identified from waiting lists in the first instance in order of referral. Recruitment will continue until SPEAKS therapists no longer have available spaces and will recommence when further spaces come available. Recruitment ended in February 2021. This was after journal submission, but prior to completion of peer reviews.

## METHODS: DATA COLLECTION, MANAGEMENT AND ANALYSIS
### Allocation, allocation concealment and blinding

Not applicable due to the single arm design.

### Data collection methods and retention

Informed consent and data at all time-points will be organised and collected by a research assistant specifically employed to complete this role. Mutually agreed appointment times for completion of all measures will be agreed. The research assistant will maintain engagement of participants with regular newsletters and individual correspondence.

### Data management and availability

Data entry errors will be checked by double entering 10% of data. Examining data for impossible values by looking at data ranges will test data quality. No post-treatment data will be released until the database is locked. Access to the final trial dataset will be available to the PIs and research assistant. The datasets generated and/or analysed during the current study will be included in the subsequent results publication. Anonymised participant-level data available on request.

### Data analysis

The data will be analysed using password protected NHS computers at NELFT. It will be analysed by PIs, the research worker and clinical psychology trainees as part of their doctoral theses. Thematic analysis of acceptability interviews will be completed before quantitative analysis of acceptability data to avoid bias. Dependent t-tests (or non-parametric equivalents) and effect sizes (Cohen's d) will be calculated for all outcomes measures. EDEQ change will be used in a power analysis to estimate sample size required for effectiveness trial. We will model for missing data of anybody who completed therapy.

## METHODS: MONITORING
### Data monitoring

A Research Steering Committee (RSG) composed of the research team including CI and PIs, representatives from the sponsors, representatives from both EDS and PPI members (with history of anorexia and family members) oversees the SPEAKS research programme. It is chaired by an experienced researcher independent from trial Trusts and sponsor. The RSG meets every 6 months for reporting and discussion. Additional meetings can be convened at short notice if required. The RSG chair is available for independent advice to the research team.

### Patient and public involvement

PPI has been in place since the inception of the study before trial funding was sought. An initial Research Design Service PPI Grant covered costs of a consultation series to assess SPEAKS model and trial validity. This informed intervention development, study design and the SPEAKS name. Following trial funding, feedback on key documents such as the plain English summary and participant information sheets (PIS) were obtained via contacts made in this initial consultation. Further, PPI input continues throughout the study via the RSG and in open feedback events at key stages of the study advertised via local charity and support group networks.

### Adverse event reporting and harms

Protocol violations such as hospital admission are recorded as per Trust guidance, and logged in trial records for reporting. Participants admitted to hospital will be withdrawn from the study. All research staff are NIHR Good Clinical Practice (GCP) trained and follow these guidelines for safety reporting procedures. Due to the feasibility design, there are no interim analyses or stopping guidelines; however, therapists monitor participant safety with regular risk assessments and communicate to the research team. Events leading to participants or others experiencing potential or actual serious harm are recorded by the researcher and reported to the PI and sponsor within 24 hours of knowledge of the event. Decision on expectedness and relatedness to the study intervention will be taken, with further investigation as required. The PI and sponsor will monitor events in case a pattern emerges, taking action if necessary. Clinical risk will be managed within ED service guidelines. Following completion of trial participation, all usual clinical service and NHS Trust SOP continue to apply with post-trial care continuing if clinically indicated. During the trial all usual Trust complaint procedures can be followed.

### Auditing

Overall study conduct, and conduct at individual sites, will be monitored by the Sponsor Monitor at regular intervals. The Trial Master File (TMF) will be audited at site initiation, annually for the duration of the study and at study closure. This audit will follow the sponsor SOP and includes checks of: completeness and secure storage

of TMF, study-wide approvals, study-wide safety and deviation/violation reporting, GCP compliance and performance of study against recruitment targets.

Individual research sites will be monitored after five participants have been recruited as part of site initiation, annually from site initiation date and at study closure at site. Monitoring will follow the sponsor SOP and include monitoring of the investigator site file for completeness and secure storage, SDV checks, CRF checks, serious adverse events and deviation/violation checks, GCP compliance and performance of site against recruitment targets.

## Ethics and dissemination
### Research ethics approval
This study has been reviewed in accordance with the guidelines for Canterbury Christ Church University research and has been approved by the London–Bromley Research Ethics Committee (NHS Rec Reference REC Ref: 19/LO/1530).

### Protocol amendments
Any protocol amendments will be communicated to all relevant parties and research sites. Notice of intention to submit an amendment will be provided via email to the sponsor and the research governance representatives at all sites to offer an opportunity to discuss any queries prior to submission and confirm support. Ethical approval will subsequently be sought via online submission following IRAS guidance. Once ethical approval has been granted for an amendment this will be disseminated to all parties and official registries (such as ISRCTN) will be notified.

### Consent
Patients meeting inclusion criteria will be provided with a PIS by their assessing clinician if they are willing to hear more about the study. Informed consent will be obtained at a face-to-face meeting with a member of the research team. The meeting will take place at least 48 hours after the participant has been provided with the PIS, with additional time given as necessary.

At the consent appointment, the research worker will answer any initial questions from the potential participant. They will review the PIS, highlighting key aspects and checking the patient's understanding of what study involvement entails. The researcher will explicitly state and make clear that deciding not to participate will not affect the patient's care in any way, and that if they decide to take part they can change their mind at any time without affecting their current or future care. The research team (like all other clinical staff members) will be trained in the principles of mental capacity and will hold this in mind throughout. Obtaining consent will be a focus of research supervision.

### Confidentiality
In line with usual clinical practice, all participant information will be kept confidential, except as governed by law (ie, if there is a legal obligation on the researcher to disclose this information to authorities due to risk concerns).

Signed informed consent forms will be stored separately from completed questionnaires and interview transcripts, which will be notated with only a participant number and/or pseudonym. Signed informed consent forms will be kept in a locked cabinet on NELFT or SPFT Property as relevant. Completed questionnaire data will be entered into study databases in encrypted Trust network folders accessible only to the research team. Any video recordings of therapy sessions and formulation descriptions will be stored on secure encrypted password protected NHS hard drives.

Participant identity will not be included in written interview transcripts, and will not be revealed in any publication resulting from this study. Data gathered from this study will be retained as required by regulations, which is up to 10 years following publication of empirical articles or communications describing study results.

## Availability of data and materials
Study databases and TMFs will only be available to the research team employed by the participating Trusts. The trainee psychologists analysing video recordings and psychological formulation data will not know participant identity beyond what they see in recorded therapy sessions. Qualitative interview recordings may be transcribed by a third-party agency approved by the sponsor. Participant names and personal details will not be disclosed and full confidentiality agreements will be established as per this protocol.

## Dissemination policy
Data will be disseminated via high-impact, peer-reviewed journals, with Open Access sought where possible. Papers will be submitted for conference presentations to achieve dissemination to practitioners and professionals within the ED field. Dissemination to service users and clinical networks is considered extremely important and be achieved via service user networks across NHS Trusts and Higher Education Institutions (HEIs) involved in the study, via charity networks (eg, UK eating disorder charity BEAT) and through a free open conference hosted by the NHS Trusts and HEI to disseminate findings and facilitate feedback on the SPEAKS clinical model and intervention. The decisions of when and where to publish data and authorship eligibility is decided by the SPEAKS RSG. Professional writers will not be used.

## DISCUSSION
This SPEAKS feasibility trial aims to assess acceptability, research and recruitment, sample size estimation and initial economic indications of the SPEAKS intervention for adults with AN Otherwise Specified Feeding or Eating Disorder-AN type. It also tests the hypothesised change process of the SPEAKS intervention.

## Potential implications

This first test of the SPEAKS intervention will provide initial indication of the feasibility of SPEAKS and inform whether progression to larger trials is appropriate. Change process analyses will provide valuable insights into SPEAKS hypotheses, informing refinement of the intervention, while also contributing to broader evidence-base of relevant change processes for adults with AN.

## Strengths

This feasibility trial is strengthened by multi-site design. Furthermore, sites included are relatively research naïve in delivery of clinical trials facilitating realistic insights into the feasibility of intervention delivery within naturalistic NHS services. Inclusion of consecutive referrals enables broad spectrum of participants across AN and OSFED severity and presentations, increasing generalisability. Robust measures to facilitate adherence to the model are included.

## Challenges

SPEAKS is an experiential, relational model, designed for face-to-face settings. This trial began in the wake of the COVID-19 pandemic resulting in therapy sessions moving indefinitely to online video platforms with most participants never meeting their therapist in person. People with AN are often regarded as 'difficult to engage' and have high levels of therapy dropout, thus issues of engagement with clinical and research protocols are a challenge even without this context. These factors may affect outcomes or length of delivery.

**Author affiliations**
¹Kent and Medway All Age Eating Disorder Service, North East London NHS Foundation Trust, Kent, UK
²Salomons Institute for Applied Psychology, Canterbury Christ Church University, Tunbridge Wells, UK
³Brighton and Hove Eating Disorder Service, Sussex Partnership NHS Foundation Trust, Brighton, UK

**Contributors** AO is the chief investigator and corresponding author. She led protocol development and wrote all trial documents. She is codeveloper of the SPEAKS intervention, and wrote the SPEAKS intervention guidebook. She is principal investigator (PI) for the Kent site. TL is codeveloper of the SPEAKS intervention and offered feedback and edits on the SPEAKS intervention guidebook. He contributed to the study design and development of the protocol. He is chair of the RSG. RB is research assistant on the SPEAKS feasibility trial. He has provided input into the protocol and assisted in the preparation and content of this manuscript. HS is codeveloper of the SPEAKS intervention and provided content, feedback and edits on the SPEAKS intervention guidebook. She contributed to the study design and development of the protocol. She is PI for the Sussex site.All authors read and approved the final manuscript.

**Funding** This feasibility study is funded as part of an Integrated Clinical Academic Fellowship-Clinical Lectureship awarded to AO (ICA-CL-2015-01-005) supported by the National Institute for Health Research and Health Education England.

**Disclaimer** The views expressed in this publication are those of the authors and not necessarily those of the NHS, the National Institute for Health Research, Health Education England or the Department of Health.Roles and Responsibilities.

**Competing interests** None declared.

**Patient and public involvement** Patients and/or the public were involved in the design, or conduct, or reporting, or dissemination plans of this research. Refer to the Methods section for further details.

**Patient consent for publication** Not applicable.

**Provenance and peer review** Not commissioned; externally peer reviewed.

**ORCID iD**
Anna Oldershaw http://orcid.org/0000-0002-9473-5715

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
