## [Reviewer comments · BMJ Open]

ARTICLE DETAILS

TITLE (PROVISIONAL)	The SPEAKS study: Study protocol of a multisite feasibility trial of the Specialist Psychotherapy with Emotion for Anorexia in Kent and Sussex (SPEAKS) intervention for outpatients with anorexia nervosa or otherwise specified feeding and eating disorders, anorexia nervosa type
AUTHORS	Oldershaw, Anna; Lavender, Tony; Basra, Randeep; Startup, Helen

VERSION 1 – REVIEW

REVIEWER	Strodl, Esben Queensland University of Technology
REVIEW RETURNED	22-Apr-2021

GENERAL COMMENTS	There is certainly a need for feasibility studies of novel psychotherapies for Anorexia Nervosa. Examining a psychotherapy that has a focus on emotional change sounds promising and so the study has scientific merit. Overall, the methodology appears to be consistent with the aims and hypotheses of the study. The manuscript appears to address the items on the SPIRIT Checklist. My main concern is that at times the description of the methodology appeared to be a little vague. I think the manuscript has the potential for being a very useful publication but can be slightly improved by adding more clarity to parts of the manuscript. I have listed below some aspects of the manuscript which I think could benefit from a little more clarity. - First sentence requires a reference.- Under the Objectives subheading, it was not clear to me what the authors meant by “validity” when they referred to “validity and acceptability” of the treatment? Validity is not referred to in the hypotheses.- Associated with this, the authors refer to this study as a feasibility study however it would help the reader if it was made more explicit in the objectives and hypotheses how the authors are defining/measuring feasibility. Will this be investigated by measuring reach and recruitment? I consider feasibility and acceptability as being separate constructs, so I presume feasibility is not measured by acceptability/fidelity/compliance – but good to make this explicit if this is the case.- I did not understand what the following hypothesis meant “the SPEAKS intervention will follow our hypothesised change process”.- Under eligibility the authors have a typo, referring to DSM V rather than DSM 5.- The exclusion criteria are a little vague at times e.g. what to do the authors mean by considerable physical risk (is risk of suicide a
---

	physical risk or a psychological risk), or comorbidities requiring treatment priority? How is alcohol/substance dependence measured and do the authors mean meeting criteria for an alcohol/substance use disorder or are they only interested in dependence? There appears to be another typo of “intellectual disabilities” rather than “intellectual disability”. Associated with this, do the authors mean a documented diagnosis of an intellectual disability or else how will this be assessed?  - The intervention is being conducted at two different locations. The authors state “All other usual care procedures (e.g. dietician appointments; carer’s workshops) will remain, but be reported.” It would be helpful for a reader to know how similar/dissimilar the other usual care procedures are between the two locations. - I read it a few times, but I did not understand the following sentence or how it related to the surrounding sentences “This diagrammatically represents the intervention, depicting anticipated intervention components, expected mechanisms of change and outcomes.” - The description of the intervention was vague. It appears to be based upon EFT and SFT, but it is not clear what the specific goals of the therapy are, what the strategies are, or what the structure of the therapy is? - The explanation of treatment fidelity was also a little vague. More information about the treatment fidelity checklist would be helpful? Who judges fidelity and upon what criteria? Will every session be video recorded and observed or a subset? If a subset then is this a random sampling or are samples from certain phases of the therapy taken? Are the entire video recordings observed or just parts of the video and if parts then how long and how are the parts viewed determined? Such information helps the reader to get a clearer view of how confident they can be about the fidelity of the treatment. - The description of the measurement of validity and acceptability could be clearer. I am still not sure what the authors mean by “validity” of the therapy. The authors will include a qualitative approach to measure “validity” and acceptability via a semi-structured interview but they do not provide the questions, or even examples of the questions, that they plan to use. As such it is not possible to gauge how appropriate the questions are. Similarly, the authors state that they will assess validity and acceptability quantitatively using analogue scales, but they do not include the questions, or types of questions, that they plan to use or the scale they will ask the participants to respond on? Are these published questions and if so it would be good to include the references in addition to a description of the questions? If they are not published, then it would still be good to have include a description of the questions as well as a justification for including them. - Associated with this, I would have thought that the authors would also measure drop-out rates as an indicator of acceptability. - The authors state “As described above, session recordings will enable assessment of adherence to SPEAKS, where consented to by participants.” However, it is not clear how adherence will be measured? - The therapeutic change process will be investigated by using two coding systems to code video recordings. However, it was not clear to me how this data will be analysed in order to examine the therapeutic process? - Associated with this, no statistical analysis plan is given.
--	--

	Overall, it looks like a great study that will provide useful preliminary evidence for the proposed novel psychotherapy. I hope that the above suggestions are helpful to the authors.
--	--

REVIEWER	Haun, Markus Heidelberg University, Department of General Internal Medicine and Psychosomatics
-----------------	---

REVIEW RETURNED	31-May-2021
-------------

GENERAL COMMENTS	Thank you for giving me the opportunity to review this interesting protocol for the feasibility trial of a specialist psychotherapy intervention for patients with anorexia nervosa. General remark: This manuscript reports a study for which recruitment has already ended. I assume that some patients have already received the intervention at least to some degree. To some extent, it will no be too late for any major (or even minor) protocol modifications. These circumstances must be stated in the paper. Please provide a SPIRIT checklist referring to the pages of your manuscript for the individual items of the SPIRIT checklist. Please provide a study flowchart and, most importantly, a study schedule. The study features a carefully conceptualized intervention. In this regard, I was wondering why the authors did not chose to run a randomised trial which would provide a sound basis for running a future larger effectiveness trial. This single arm study does not allow for estimating the acceptability of a randomised design which is a major limitation. Moreover, this seems to be the third version of the protocol. Please describe any changes made to the original protocol in the manuscript. Abstract: Please indicate the exact sample size for the trial and add the trial design (multi-site single arm within-group mixed-methods design). Concerning the outcomes, I would suggest measuring the acceptability by calculating the rate of participants who will complete the intervention. It may be worthwhile contemplating to include recruitment yield (do you mean recruitment yield by writing "conversion rate"?) and follow-up rates. At any rate, I would measure adverse events related to the trial and/or intervention. Please explain what BEAT and HEI refer to. Introduction: Well done. However, I would encourage the authors to add the specific aim for investigating sample sizes – what is the long-term goal? Will there be an efficacy/effectiveness trial with piggy-back economic evaluation? It is not clear to the reader what this feasibility trial should prepare for. Methods: Drops of more than 500 grams in a week are quite frequent in patients with anorexia nervosa. Please specify this criterion. For each eligibility criterion, please indicate how you will assess it. Please use the TIDieR framework/checklist for describing the SPEAKS intervention. Given the population of the therapists, please describe how you will deal with potential allegiance effects. Please drop the confidence interval approach for sample size calculation and replace it with simple recommendations, e.g., by the NIHR (https://www.rds-london.nihr.ac.uk/RDSLONDON/media/RDSCONTENT/files/PDFs/Justifying-SampleSize-for-a-Feasibility-Study_1.pdf) – its use has been discouraged previously.
--

VERSION 1 – AUTHOR RESPONSE

Reviewer: 1 - Dr. Esben Strodl, Queensland University of Technology

- First sentence requires a reference.

Thank you. We have added the following reference.

Smink FR, Van Hoeken D, Hoek HW. Epidemiology of eating disorders: incidence, prevalence and mortality rates. *Current psychiatry reports*. 2012 Aug;14(4):406-14.

- Under the Objectives subheading, it was not clear to me what the authors meant by “validity” when they referred to “validity and acceptability” of the treatment? Validity is not referred to in the hypotheses. Associated with this, the authors refer to this study as a feasibility study however it would help the reader if it was made more explicit in the objectives and hypotheses how the authors are defining/measuring feasibility. Will this be investigated by measuring reach and recruitment? I consider feasibility and acceptability as being separate constructs, so I presume feasibility is not measured by acceptability/fidelity/compliance – but good to make this explicit if this is the case.

We have removed the reference to validity as we can see this is confusing. We wanted to complete a feasibility study to investigate whether a trial will be feasible, to determine “can it be done?” This is largely measured by reach and recruitment, but also by acceptability of the novel intervention (including whether people drop out of therapy) and also whether therapists were able to appropriately adhere to its delivery as intended. We also wanted to analyse and confirm the hypothesised change process in order for the therapy model to be appropriately updated before progressing to a full effectiveness trial (if it is indicated).

- I did not understand what the following hypothesis meant “the SPEAKS intervention will follow our hypothesised change process”.

We apologise, we can see this sentence doesn’t make sense. We have now reworded the sentence and have also added a Figure which illustrates this hypothesis (please see Figure 1 and hypothesis section page 7).

- Under eligibility the authors have a typo, referring to DSM V rather than DSM 5.

This has been corrected, thank you.

- The exclusion criteria are a little vague at times e.g. what do the authors mean by considerable physical risk (is risk of suicide a physical risk or a psychological risk), or comorbidities requiring treatment priority?

We apologise that this was not well explained. This relates to eating disorder risk. We use an assessment in the UK called MARSIPAN, which largely incorporates physical risk factors, but also includes some psychological risk factors such as suicide risk. We excluded those who rated high risk or high concern on the MARSIPAN rating, particularly pertaining to their physical risk in deteriorating ED health (e.g. weight loss). We have reworded this criterion and included appropriate references. We have also clarified by separating out physical and psychological risk. Please see page 8.

- How is alcohol/substance dependence measured and do the authors mean meeting criteria for an alcohol/substance use disorder or are they only interested in dependence?

This means meeting criteria for alcohol/substance use disorder and has now been changed to be clear (page 8).

- There appears to be another typo of “intellectual disabilities” rather than “intellectual disability”. Associated with this, do the authors mean a documented diagnosis of an intellectual disability or else how will this be assessed?

This typo has been corrected and detail added to indicate that this would be a diagnosed intellectual disability impeding use of therapy (page 8).

- The intervention is being conducted at two different locations. The authors state “All other usual care procedures (e.g. dietician appointments; carer’s workshops) will remain, but be reported.” It would be helpful for a reader to know how similar/dissimilar the other usual care procedures are between the two locations.

The two services included in the trial are commissioned slightly differently; however usual care procedures are similar across both for those receiving psychological therapy because they both follow national NHS guidelines on treatment for eating disorders. We have added this information (page 9).

- I read it a few times, but I did not understand the following sentence or how it related to the surrounding sentences “This diagrammatically represents the intervention, depicting anticipated intervention components, expected mechanisms of change and outcomes.”

Sorry this wasn’t clear. This has been reworded as follows and a Figure added. We hope this aids readability and clarity (page 10).

SPEAKS is organised into five phases, with associated mechanisms of change and therapeutic tasks described for each phase (see Figure 2).

- The description of the intervention was vague. It appears to be based upon EFT and SFT, but it is not clear what the specific goals of the therapy are, what the strategies are, or what the structure of the therapy is?

We hope that the two new Figures provided (Figures 1 and 2) provide a helpful description of the intervention for the purposes of this paper. Further to Dr Haun’s review below, we have also utilised the TIDieR framework to ensure we have included key intervention information and we have included the completed checklist in supplementary data.

- The explanation of treatment fidelity was also a little vague. More information about the treatment fidelity checklist would be helpful? Who judges fidelity and upon what criteria? Will every session be video recorded and observed or a subset? If a subset then is this a random sampling or are samples from certain phases of the therapy taken? Are the entire video recordings observed or just parts of the video and if parts then how long and how are the parts viewed determined? Such information helps the reader to get a clearer view of how confident they can be about the fidelity of the treatment.

The treatment fidelity checklist includes ensuring that certain procedures are in place, such as those listed in the text, e.g. utilising a guidebook/manual and standardised training for therapists. All sessions are recorded and a random selection of therapy tapes will also be assessed for fidelity and adherence to the model by the intervention developers. They will assess core components of the treatments such as empathy (using the Therapist Empathy Scale; Decker et al., 2014), appropriate task selection based on identified tasks markers, and appropriate task resolution. This detail and reference have been added to the paper (page 12).

- The description of the measurement of validity and acceptability could be clearer. I am still not sure what the authors mean by “validity” of the therapy.

We have removed the reference to validity as we can see this is confusing.

- The authors will include a qualitative approach to measure “validity” and acceptability via a semi-structured interview but they do not provide the questions, or even examples of the questions, that they plan to use. As such it is not possible to gauge how appropriate the questions are. Similarly, the authors state that they will assess validity and acceptability quantitatively using analogue scales, but they do not include the questions, or types of questions, that they plan to use or the scale they will ask the participants to respond on? Are these published questions and if so it would be good to include the references in addition to a description of the questions? If they are not published, then it would still be good to have include a description of the questions as well as a justification for including them.

Thank you for highlighting this omission. The qualitative interview is a slightly adapted version of the Elliott Client Change Interview Schedule. This interview schedule asks about the participant's experience of therapy, valued and unhelpful aspects of the intervention as well as broader questions about the acceptability of the research trial. The VAS used is not a published scale, but was a quantitative measure of some of the questions asked in the qualitative interview on a seven point scale from strongly agree to strongly disagree. Additional detail pertaining to these measures has been added (page 11).

- Associated with this, I would have thought that the authors would also measure drop-out rates as an indicator of acceptability.

Yes we are measuring this. Apologies that this was not clearly stated in the paper; it has been added (page 11).

- The authors state “As described above, session recordings will enable assessment of adherence to SPEAKS, where consented to by participants.” However, it is not clear how adherence will be measured?

We hope that our answer to previous request for clarification on fidelity and adherence also answers this point.

- The therapeutic change process will be investigated by using two coding systems to code video recordings. However, it was not clear to me how this data will be analysed in order to examine the therapeutic process?

We have added in some additional detail (pages 12-13). Due to the constraints of the word count we have needed to keep it brief, but hope it is now sufficiently clear.

- Associated with this, no statistical analysis plan is given.

We have added in brief details of our analysis plan within what the word count would allow and hope it provides sufficient indication of what we plan to do (page 15).

Reviewer: 2 - Dr. Markus Haun, Heidelberg University

- General remark: This manuscript reports a study for which recruitment has already ended. I assume that some patients have already received the intervention at least to some degree. To some extent, it

will not be too late for any major (or even minor) protocol modifications. These circumstances must be stated in the paper.

We have added this information to the protocol in the Recruitment section (page 14).

“Recruitment ended in February 2021. This was after submission to the current journal, but prior to completion of peers reviews.”

- Please provide a SPIRIT checklist referring to the pages of your manuscript for the individual items of the SPIRIT checklist. Please provide a study flowchart and, most importantly, a study schedule.

A completed SPIRIT checklist, a completed TIDieR checklist, study flowchart and study schedule have all been submitted.

- The study features a carefully conceptualized intervention. In this regard, I was wondering why the authors did not choose to run a randomised trial which would provide a sound basis for running a future larger effectiveness trial. This single arm study does not allow for estimating the acceptability of a randomised design which is a major limitation.

At this stage in the intervention development, we felt it was inappropriate to advance quickly to a randomised design in terms of both the ethics and costs associated. We felt we needed to assess ‘can it be done?’ by delivering SPEAKS in the services to address factors pertaining to implementation and treatment viability/length. There are however questions asked regarding acceptability of a randomised controlled trial in the qualitative acceptability interview completed with all participants. Questions include willingness to be randomised and acceptability of questionnaires that were used in order to inform a full trial protocol. We have added this detail to the paper in the Outcomes section (page 11).

- Moreover, this seems to be the third version of the protocol. Please describe any changes made to the original protocol in the manuscript.

This paper is the third version because several changes were made due to the COVID-19 pandemic which occurred following the original protocol. A further version was created due to the addition of the two change process analyses. This has been described in the ‘Protocol Version’ section (page 4).

- Abstract: Please indicate the exact sample size for the trial and add the trial design (multi-site single arm within-group mixed-methods design).

We have added the trial design in the abstract, thank you. Unfortunately, we don’t yet know the final sample size of people completing as not everybody has completed therapy yet.

- Concerning the outcomes, I would suggest measuring the acceptability by calculating the rate of participants who will complete the intervention. It may be worthwhile contemplating to include recruitment yield (do you mean recruitment yield by writing “conversion rate”?) and follow-up rates. At any rate, I would measure adverse events related to the trial and/or intervention.

We do measure all of these and apologise for not stating this more clearly (see pages 11, 16, 17). We mean recruitment yield by “conversion rate” and have changed this description throughout.

- Please explain what BEAT and HEI refer to.

BEAT is an eating disorder charity in the UK and HEI refers to Higher Education Institution (University). We have now added this detail to the paper.

- Introduction: Well done. However, I would encourage the authors to add the specific aim for investigating sample sizes – what is the long-term goal? Will there be an efficacy/effectiveness trial with piggy-back economic evaluation? It is not clear to the reader what this feasibility trial should prepare for.

This has been added to the Abstract and Objectives, thank you.

- Methods: Drops of more than 500 grams in a week are quite frequent in patients with anorexia nervosa. Please specify this criterion. For each eligibility criterion, please indicate how you will assess it.

On the basis of your feedback and that of the other reviewer we have clarified the eligibility criteria, with reference to the MARSIPAN guidelines which are used to assess risk in patients in with anorexia nervosa in the UK and is our standard process for all patients in these services (page 8).

- Please use the TIDieR framework/checklist for describing the SPEAKS intervention.

Thank you for drawing our attention to this framework and checklist. These have now been used and a completed TIDieR checklist added to the supplementary materials.

- Given the population of the therapists, please describe how you will deal with potential allegiance effects.

In order to minimise the possibility of potential allegiance effects, an independent Research Assistant is employed to collect informed consent and collect all outcome data. This has been made clearer, but also added to the Limitations (page 3).

- Please drop the confidence interval approach for sample size calculation and replace it with simple recommendations, e.g., by the NIHR (https://www.rds-london.nihr.ac.uk/RDSLONDON/media/RDSContent/files/PDFs/Justifying-SampleSize-for-a-Feasibility-Study_1.pdf) – its use has been discouraged previously.

Thank you for highlighting this. The sample size was calculated as advised by the NIHR RDS in 2015 when the study was first conceptualised for the funding application of the SPEAKS programme (of which this feasibility trial is one part). This is the sample size for which ethical approval was subsequently obtained and is what we have been allocated funding to achieve. Utilising the approach suggested, we can say with a sample size of 36, we will be able to estimate a drop-out rate of 40% to within a 95% confidence interval of +/- 16%. Please advise us whether this is what you would like us to replace the text with.

VERSION 2 – REVIEW

REVIEWER	Strodl, Esben Queensland University of Technology
REVIEW RETURNED	12-Oct-2021

GENERAL COMMENTS	I think the authors have addressed the questions I had regarding this manuscript. It is an interesting study and will guide the development of larger future studies. While I think the manuscript
--

	is OK to be published as is, a couple of minor suggestions for the authors to consider are: 1) making it more explicit how the measures of economic evaluation fits in with the aims and hypotheses of the study; 2) making it more explicit in the data analysis section how the CAMS data will be analysed for "temporal patterns in the expression of emotion codes over time" and how this will help test hypothesis 4. I look forward to reading about the completed study in due time.
--	--

REVIEWER	Haun, Markus Heidelberg University, Department of General Internal Medicine and Psychosomatics
REVIEW RETURNED	25-Sep-2021

GENERAL COMMENTS	Thank you for considering my suggestions from the previous round. I think this sound study protocol ready for publication. One last minor point regarding your response ". The sample size was calculated as advised by the NIHR RDS in 2015 when the study was first conceptualised for the funding application of the SPEAKS programme (of which this feasibility trial is one part). This is the sample size for which ethical approval was subsequently obtained and is what we have been allocated funding to achieve. Utilising the approach suggested, we can say with a sample size of 36, we will be able to estimate a drop-out rate of 40% to within a 95% confidence interval of +/- 16%. Please advise us whether this is what you would like us to replace the text with.": Thank you for clarifying this issue. I would be grateful if you amended the manuscript accordingly.
--

VERSION 2 – AUTHOR RESPONSE

Reviewer: 1

Dr. Esben Strodl, Queensland University of Technology

Comments to the Author:

I think the authors have addressed the questions I had regarding this manuscript. It is an interesting study and will guide the development of larger future studies. While I think the manuscript is OK to be published as is, a couple of minor suggestions for the authors to consider are:

1) making it more explicit how the measures of economic evaluation fits in with the aims and hypotheses of the study;

The measures of economic evaluation are an added factor to the feasibility design. We want to have some estimation of costs such that we can establish whether the intervention is economically feasible. If the intervention is too costly for the NHS or other institutions to implement on a wider scale, then proceeding to a larger randomised controlled trial (RCT) would not be indicated. If costs don't prohibit proceeding to an RCT, the economic evaluation will assist in establishing required RCT funding for a grant application. We have added the following sentence to the Objectives on page 6:

- Sample size & economic evaluation to establish parameters and financial costs of a potential future efficacy/effectiveness trial

2) making it more explicit in the data analysis section how the CAMS data will be analysed for "temporal patterns in the expression of emotion codes over time" and how this will help test hypothesis 4.

Thank you, there is lot more we would have liked to have said here, but we were very restricted by the word count. I have tweaked the wording in the hope that meaning is now clearer and have also provided an additional reference which uses the same form of statistical analysis (page 13, paragraph 1).

I look forward to reading about the completed study in due time.

Thank you for your positive and helpful comments.

Reviewer: 2

Dr. Markus Haun, Heidelberg University

Comments to the Author:

Thank you for considering my suggestions from the previous round. I think this sound study protocol ready for publication. One last minor point regarding your response ". The sample size was calculated as advised by the NIHR RDS in 2015 when the study was first conceptualised for the funding application of the SPEAKS programme (of which this feasibility trial is one part). This is the sample size for which ethical approval was subsequently obtained and is what we have been allocated funding to achieve. Utilising the approach suggested, we can say with a sample size of 36, we will be able to estimate a drop-out rate of 40% to within a 95% confidence interval of +/- 16%. Please advise us whether this is what you would like us to replace the text with.": Thank you for clarifying this issue. I would be grateful if you amended the manuscript accordingly.

Thank you for your positive and helpful comments. We have amended the manuscript with regards to sample size (page 14, paragraph 3).